# SANDWake3D: A 3D parabolic RANS solver for atmospheric boundary layers and turbine wakes

Lawrence Cheung<sup>1</sup>, Prakash Mohan<sup>2</sup>, Marc T. Henry de Frahan<sup>2</sup>, Gopal R. Yalla<sup>3</sup>, Alan Hsieh<sup>3</sup>, Kenneth Brown<sup>3</sup>, Nathaniel deVelder<sup>3</sup>, Sam Kaufman-Martin<sup>1,4</sup>, Marc Day<sup>2</sup>, and Michael Sprague<sup>2</sup>

**Correspondence:** Lawrence Cheung (lcheung@sandia.gov)

Abstract. Despite many recent advances, modeling wind turbine wakes using semi-empirical and analytical models still face challenges when dealing with more complicated situations involving wind shear, veer, atmospheric stratification, and wake superposition. To address these limitations, this study introduces a three-dimensional, parabolic Reynolds Averaged Navier Stokes (RANS) k-epsilon formulation which includes an atmospheric boundary layer model and an actuator disk model for turbine wakes. The full three-dimensional solution for the velocity, temperature, and turbulence variables are efficiently solved through an alternating direction implicit scheme that requires orders of magnitude less computational resources than traditional high fidelity approaches. The results of the parabolic RANS model are compared to the equivalent large-eddy simulations (LES) and semi-empirical wake models at different wind speeds under stable atmospheric conditions with veer and shear. For the single turbine wake the RANS model was able to capture the wake deficit behavior, including the wake stretching and skewing that was observed in the LES. The distribution of the wake turbulence in the RANS model also agreed with results from the higher fidelity simulations. In simulations of a two-turbine, directly waked configuration, the new RANS model was able to handle the wake superposition behavior without difficulty, and also correctly modeled the corresponding increase in wake turbulence when compared to LES. Lastly, a demonstration of the RANS model on a 9-turbine, 3 row wind farm is shown and compared to LES.

15 *Copyright statement.* This written work is authored by an employee of NTESS. The employee, not NTESS, owns the right, title and interest in and to the written work and is responsible for its contents.

#### 1 Introduction

The complex behavior of wind turbine wakes has led to a very rich and fruitful area of research, but also revealed a number of challenges to those developing wind farm wake models for general use. A history of measurements and simulations have shown that turbine wake behavior is influenced by a number of factors, including interactions with the shear, veer, and stratification

<sup>&</sup>lt;sup>1</sup>Sandia National Laboratories, Livermore, CA, USA

<sup>&</sup>lt;sup>2</sup>National Renewable Energy Laboratory, Golden, CO, USA

<sup>&</sup>lt;sup>3</sup>Sandia National Laboratories, Albuquerque, NM, USA

<sup>&</sup>lt;sup>4</sup>University of California, Santa Barbara, CA, USA

https://doi.org/10.5194/wes-2025-249

Preprint. Discussion started: 24 November 2025

© Author(s) 2025. CC BY 4.0 License.

25

in the atmospheric boundary layer (ABL), wake-to-wake interactions, wake steering, and the development of wake-added turbulence. High-fidelity modeling, including Large Eddy Simulations (LES), can consistently capture (Cheung et al., 2023; Hsieh et al., 2025) all of these complex behaviors, but remains too computationally expensive to be used for wind farm optimization or design purposes.

Many analytic and semi-empirical models have been developed to quickly calculate wake behavior and predict wind farm performance under a variety of conditions. Starting from the simplest Jensen model (Jensen, 1983) to more recent empirical Gaussian models (Bastankhah and Porté-Agel, 2014; Niayifar and Porté-Agel, 2016), these models typically adopt an assumed functional form for the wake profile with free parameters which are calibrated to match the wake behavior in specific scenarios. These semi-empirical models are generally combined with other models to capture the effects of wake superposition (Gunn et al., 2016) or wake-added turbulence (Crespo et al., 1996). More recent work (Narasimhan et al., 2022, 2025) has extended analytical wake models to include atmospheric shear and veer, but consistently accounting for these effects in interacting wakes or in the wake-added turbulence behavior remains an open question.

Previous studies have demonstrated the potential of parabolic Reynolds Averaged Navier Stokes (RANS) methods when compared to semi-analytic methods. Starting from the work of Ainslie (1988), who developed an axisymmetric formulation for a single turbine wake, the later work of Iungo et al. (2018) explored the use of a mixing-length eddy viscosity model for turbulent inflow. In the work of Letizia and Iungo (2022), lidar measurements were used to calibrate a depth-averaged parabolic RANS method for operational wind farms. Cheung et al. (2024) used a simplified two-dimensional RANS model to study the interaction of large-scale convective structures in an unstable ABL with wind turbine wakes. Another recent study by Cheung et al. (2025) coupled an axisymmetric RANS solution with a linear stability model to capture the development of coherent structures in turbine wakes when active wake control is applied.

Of particular interest to the current work are three-dimensional parabolic models, including the WakeBlaster model of Bradstock and Schlez (2020) and the combined curl model of Martínez-Tossas et al. (2021). In the WakeBlaster model, a single streamwise momentum equation is solved by advancing 2D planes of the velocity field, and the introduction of wakes is accomplished through direct manipulation of the velocity profiles. However this limits the ability of the model to handle veered or stratified inflow conditions.

Similarly, in the combined curl model, the velocity field is decomposed into a base and wake deficit variable, with a single streamwise momentum equation solved for the wake deficit. As mentioned in Martínez-Tossas et al. (2021), the combined curl model does not enforce continuity, and the turbine wakes are initialized through the velocity field rather than through body forces in the momentum equation itself. This restricts the ability of the model to handle effects such as wake-ABL interactions or wake skewing and veering.

To overcome these limitations of earlier models, the current study introduces an efficient three-dimensional RANS model which naturally captures complex effects such as shear, veer, atmospheric stratification, and turbine and wake turbulence superposition. This model, known as SANDWake3D, combines a parabolic  $k - \epsilon$  RANS with an atmospheric boundary layer solver and an actuator disk method for representing turbines. The full three-dimensional solution for all velocity, temperature, and turbulence variables can be quickly solved through an alternating direction implicit (ADI) scheme with minimal computational

Figure 1. Schematic showing the parabolized RANS solution process by marching planes of the velocity variable u(y,z) downstream through the domain.

resources. After calibration against high fidelity simulations, we show that the RANS method can accurately predict wake behavior in stably stratified conditions with a fraction of the cost of typical LES methods.

In the following sections, we first discuss the formulation of the parabolized RANS method and the numerical solution algorithm used in this study. The details of the wind simulations and turbine configurations are presented in Sec. 3, followed by a comparison of the RANS results with corresponding LES and FLORIS (Sinner and Fleming, 2024) models. In the final section, we conclude with a summary of the study and discuss recommendations for future work in this area.

#### 2 Formulation

#### 2.1 Parabolized RANS method

In the SANDWake3D model, an underlying RANS formulation was selected based on its ability to capture both the ABL behavior and the turbine wake dynamics. Previous studies have shown that the  $k - \epsilon$  model can accurately simulate stratified ABL conditions (Alinot and Masson, 2005), and was also successfully used in prior simplified models for wake dynamics (Cheung et al., 2025). Thus, assuming an incompressible, steady flow over flat terrain, the governing  $k - \epsilon$  RANS equations from Alinot and Masson (2005) are used as a starting point for this analysis.

These equations are further simplified by assuming that the second order derivatives in the streamwise direction x are small relative to those in the lateral y and vertical z directions. This leads to the following parabolic equations for mass conservation:

© Author(s) 2025. CC BY 4.0 License.

$$\frac{\partial u}{\partial x} + \frac{\partial v}{\partial y} + \frac{\partial w}{\partial z} = 0 \tag{1a}$$

as well as momentum conservation for the (u, v, w) mean velocities:

$$u\frac{\partial u}{\partial x} + v\frac{\partial u}{\partial y} + w\frac{\partial u}{\partial z} = -\frac{1}{\rho}\frac{\partial p}{\partial x} + \frac{\partial}{\partial y}\left((\nu + \nu_T)\frac{\partial u}{\partial y}\right) + \frac{\partial}{\partial z}\left((\nu + \nu_T)\frac{\partial u}{\partial z}\right) + f_{T,x} \tag{1b}$$

$$75 \quad u\frac{\partial v}{\partial x} + v\frac{\partial v}{\partial y} + w\frac{\partial v}{\partial z} = -\frac{1}{\rho}\frac{\partial p}{\partial y} + \frac{\partial}{\partial y}\left((\nu + \nu_T)\frac{\partial v}{\partial y}\right) + \frac{\partial}{\partial z}\left((\nu + \nu_T)\frac{\partial v}{\partial z}\right) + f_{T,y} \tag{1c}$$

$$u\frac{\partial w}{\partial x} + v\frac{\partial w}{\partial y} + w\frac{\partial w}{\partial z} = -\frac{1}{\rho}\frac{\partial p}{\partial z} + \frac{\partial}{\partial y}\left((\nu + \nu_T)\frac{\partial w}{\partial y}\right) + \frac{\partial}{\partial z}\left((\nu + \nu_T)\frac{\partial w}{\partial z}\right) + g\beta(\Theta - \Theta_0) + f_{T,z}$$
(1d)

In equations (1b-1d), p is the pressure,  $\rho$  is the fluid density,  $\nu$  is the kinematic viscosty, and  $\nu_T$  is the turbulent viscosity. The Boussinesq approximation is used to capture the effects of buoyancy, so the density is assumed to vary linearly with the potential temperature  $\Theta$  in the vertical direction. The body force,  $\mathbf{f_T}$ , in the momentum equations is used to represent the turbine rotor disk forces as described in Sec. 2.4. In equation (1d), the gravitational acceleration constant is g,  $\beta$  is the volumetric thermal expansion coefficient, and  $\Theta_0$  is reference potential temperature.

Similar parabolic equations can be written for the turbulent kinetic energy k and dissipation  $\epsilon$  variables:

$$u\frac{\partial k}{\partial x} + v\frac{\partial k}{\partial y} + w\frac{\partial k}{\partial z} = P_k - \epsilon + \frac{\partial}{\partial y}\left((\nu + \frac{\nu_T}{\sigma_k})\frac{\partial k}{\partial y}\right) + \frac{\partial}{\partial z}\left((\nu + \frac{\nu_T}{\sigma_k})\frac{\partial k}{\partial z}\right) + G_B \tag{1e}$$

85 
$$u\frac{\partial \varepsilon}{\partial x} + v\frac{\partial \varepsilon}{\partial y} + w\frac{\partial \varepsilon}{\partial z} = \frac{C_{1\varepsilon}}{\mathcal{T}} \left[ P_k + (1 - C_{3\varepsilon})G_B \right] - \frac{C_{2\varepsilon}}{\mathcal{T}} \varepsilon + \frac{\partial}{\partial y} \left( (\nu + \frac{\nu_T}{\sigma_\varepsilon}) \frac{\partial \varepsilon}{\partial y} \right) + \frac{\partial}{\partial z} \left( (\nu + \frac{\nu_T}{\sigma_\varepsilon}) \frac{\partial \varepsilon}{\partial z} \right)$$
(1f)

as well as for the potential temperature  $\Theta$ :

$$u\frac{\partial\Theta}{\partial x} + v\frac{\partial\Theta}{\partial y} + w\frac{\partial\Theta}{\partial z} = \frac{\partial}{\partial y}\frac{\nu_t}{\sigma_T}\frac{\partial\Theta}{\partial y} + \frac{\partial}{\partial z}\frac{\nu_t}{\sigma_T}\frac{\partial\Theta}{\partial z}.$$
 (1g)

In equations (1e) and (1f), the shear production term  $P_k$  is calculated from

$$P_k = \nu_t \left( \frac{\partial u_i}{\partial x_j} + \frac{\partial u_j}{\partial x_i} \right) \frac{\partial u_i}{\partial x_j},\tag{2}$$

while the buoyancy production term  $G_B$  is defined as

$$G_b = \beta g \frac{\nu_T}{\rho \sigma_T} \left( \frac{\partial \Theta}{\partial z} - \frac{g}{c_p} \right),\tag{3}$$

where  $c_p$  is the specific heat at constant pressure. The turbulent viscosity  $\nu_T$  is calculated as

$$\nu_T = C_{\mu} k T$$
,

where, following Durbin (1991), the timescale  ${\mathcal T}$  is the larger of

$$\mathcal{T} = \max\left(\frac{k}{\varepsilon}, 6\sqrt{\frac{\nu}{\varepsilon}}\right)$$

In equations (1e), (1f), (1g), the standard values for the  $\sigma_k$ ,  $\sigma_\epsilon$ , and  $\sigma_T$  coefficients (Jones and Launder, 1972) are used:

$$\sigma_k = 1.0, \quad \sigma_\epsilon = 1.3, \quad \sigma_T = 1.0.$$
 (4)

As discussed in Sec. 2.5, the values for the adjustable parameters  $C_{\mu}$ ,  $C_{1\epsilon}$ , and  $C_{2\epsilon}$ , along with an additional parameter  $C_k$ , are determined through calibration against LES data. The value for  $C_{3\epsilon}$  is dependent on the atmospheric stratification, and in this study, the same function as Alinot and Masson (2005) is used:

$$C_{3\epsilon}\left(\frac{z}{L}\right) = \sum_{n=0}^{5} A_n \left(\frac{z}{L}\right)^n \tag{5}$$

where L is the Monin-Obukhov length and the values of  $A_n$  are given in table 1 of Alinot and Masson (2005).

The structure of the parabolic equations (1) allows for an efficient solution algorithm to be constructed that accurately captures the three-dimensional behavior of turbine wakes. Starting from a given inflow profile  $u_0(x_0, y, z)$  at an initial streamwise position  $x_0$ , the solution planes u(x, y, z) for downstream locations can be determined through an implicit marching process (see Fig. 1). Details on the numerical solution algorithm are discussed below in Sec. 2.3.

# 2.1.1 Pressure Poisson equation

In the parabolic formulation, enforcing continuity (1a) is possible by developing the appropriate pressure Poisson equation.

Taking the divergence of the momentum equations (1b-1d), and applying the continuity equation (1a) leads to the following equation for pressure:

$$\frac{1}{\rho} \left( \frac{\partial^2 p}{\partial y^2} + \frac{\partial^2 p}{\partial z^2} \right) = -\frac{\partial u_j}{\partial x_i} \frac{\partial u_i}{\partial x_j} + g\beta \frac{\partial \Theta}{\partial z} 
+ \frac{\partial \nu_T}{\partial x_i} \left( \frac{\partial^2 u_i}{\partial y^2} + \frac{\partial^2 u_i}{\partial z^2} \right) + \frac{\partial^2 \nu_T}{\partial y \partial x_i} \frac{\partial u_i}{\partial y} + \frac{\partial^2 \nu_T}{\partial z \partial x_i} \frac{\partial u_i}{\partial z} 
= S_p$$
(6)

Note that the second derivative of pressure,  $\partial^2 p/\partial x^2$ , is neglected in the left-hand side of (6) as changes in the streamwise direction are assumed to be small relative to the y and z directions.

While many efficient algorithms exist to solve the two-dimensional Poisson problem, equation (6) can be reformulated as a parabolic diffusion problem if we assume that the pressure also depends on an artificial time  $\tau$  variable such that

$$\frac{\partial p}{\partial \tau} = -\left(\frac{\partial^2 p}{\partial y^2} + \frac{\partial^2 p}{\partial z^2}\right) + \rho S_p \tag{7}$$

As the solution to equation (7) reaches steady state, where  $\partial p/\partial \tau \to 0$ , we see that the pressure also satisfies the original equation (6). However, the same solution algorithm used to solve equations (1b-1g) can also be applied to (7), which simplifies the overall implementation as described in Sec. 2.3.

# 2.2 Inflow and boundary conditions

Following Alinot and Masson (2005), the inflow conditions to the RANS model are based on Monin-Obukhov similarity theory for thermally stratified atmospheric boundary layers over uniform flat terrain. When combined with the parabolic formulation in equations (1), this leads to a consistent approach for handling the effects of stratification in both the wake and background inflow. In this formulation, the Monin-Obukhov length L is calculated as

$$L = \frac{u_*^2 T_w}{\kappa g T_*} \tag{8}$$

where g is the gravitational constant,  $\kappa$  is the von Karman constant,  $T_w$  is the wall temperature, the friction velocity  $u_* = \sqrt{\tau_w/\rho}$  for a given wall shear stress  $\tau_w$ , the temperature  $T_* = -q_w/(\rho c_p u_*)$ , the surface heat flux  $q_w$ , the heat capacity  $c_p$ . The non-dimensional wind shear  $\phi_m$  is expressed as

$$\phi_m\left(\frac{z}{L}\right) = \begin{cases} \left(1 - 16\frac{z}{L}\right)^{-1/4}, & L 

The initial kinetic energy and dissipation profiles matched those used in Alinot and Masson (2005)

$$k_0(z) = 5.48C_k u_*^2 \sqrt{\frac{\phi_\epsilon(z/L)}{\phi_m(z/L)}},$$
 (13)

$$\epsilon_0(z) = \frac{u_*^3}{\kappa z} \phi_{\epsilon}(z/L),\tag{14}$$

where  $C_k$  is an adjustable parameter.

At the lower boundary  $z = z_{min}$ , the following Dirichlet boundary conditions are imposed:

$$u = u_0(z_{min}), \quad v = v_0(z_{min}), \quad w(z_{min}) = 0, \quad \Theta = \Theta_0(z_{min}), \quad k = k_0(z_{min}), \quad \epsilon = \epsilon_0(z_{min}), \quad p = 0,$$
 (15)

while at the upper boundary  $z = z_{max}$ , a combination of Dirichlet and Neumann boundary conditions are used:

$$u = u_0(z_{max}), \quad v = v_0(z_{max}), \quad \frac{\partial w}{\partial z} = 0, \quad \frac{\partial \Theta}{\partial z} = \Gamma, \quad k = k_0(z_{max}), \quad \frac{\partial \epsilon}{\partial z} = 0, \quad \frac{\partial p}{\partial z} = 0$$
 (16)

where  $\Gamma$  is the specified lapse rate.

#### 145 2.3 Numerical solution


The numerical solution to equations (1) is computed using an iterative alternating-direction implicit (ADI) approach. This allows for an efficient and robust marching procedure by splitting the y and z differentiations into separate stages which can be quickly calculated using a tri-diagonal matrix solver. The implicit nature of the algorithm also allows for relatively large  $\Delta x$  steps in the streamwise direction. Using the notation  $u(x,y,z)=U^n_{ij}$  to indicate the discretized flow variables at the location  $(x_n,y_i,z_j)$ , then the following second order differentiation stencils for the first and second derivatives in the y and z directions can be written as

$$D_y U_{ij} = \frac{U_{i+1,j} - U_{i-1,j}}{2}, \qquad D_y^2 U_{ij} = U_{i+1,j} - 2U_{i,j} + U_{i-1,j}$$
(17a)

$$D_z U_{ij} = \frac{U_{i,j+1} - U_{i,j-1}}{2}, \qquad D_z^2 U_{ij} = U_{i,j+1} - 2U_{i,j} + U_{i,j-1}$$
(17b)

In the advection terms of equations (1), the averaged values of the velocities and turbulent viscosity between position  $x_n$  and  $x_{n+1}$  are used:

$$\tilde{U}_{ij} = \frac{U_{ij}^{n+1} + U_{ij}^n}{2}, \qquad \tilde{V}_{ij} = \frac{V_{ij}^{n+1} + V_{ij}^n}{2}, \qquad \tilde{W}_{ij} = \frac{W_{ij}^{n+1} + W_{ij}^n}{2}, \qquad \tilde{\nu}_{T,ij} = \frac{\nu_{T,ij}^{n+1} + \nu_{T,ij}^n}{2}.$$

Solving each of the equations (1b-1g) and (7) uses a two-stage process, where advancing from  $x_n$  to  $x_{n+1}$  requires two half-steps of size  $\Delta x/2$  each. Taking the solution of the x momentum equation (1b) as an example, the first half step solves for  $U_{ij}^{n+1/2}$  given  $U_{ij}^n$  by treating the y-direction implicitly and the z-direction explicitly:

$$\left[\frac{\tilde{U}_{ij}}{\Delta x/2} + \left(\tilde{V}_{ij} - \frac{D_y \tilde{\nu}_T}{\Delta y}\right) \frac{D_y}{\Delta y} - (\nu + \tilde{\nu}_T) \frac{D_y^2}{(\Delta y)^2}\right] U_{ij}^{n+1/2} = 
\left[\frac{\tilde{U}_{ij}}{\Delta x/2} - \left(\tilde{W}_{ij} - \frac{D_z \tilde{\nu}_T}{\Delta z}\right) \frac{D_z}{\Delta z} + (\nu + \tilde{\nu}_T) \frac{D_z^2}{(\Delta z)^2}\right] U_{ij}^n + f_x^n - \frac{p_{ij}^{n+1} - p_{ij}^n}{\rho \Delta x} \tag{18a}$$

The second half-step then advances the solution from  $U_{ij}^{n+1/2}$  to  $U_{ij}^n$  by treating the z-direction implicitly and the y-direction explicitly:

$$\left[\frac{\tilde{U}_{ij}}{\Delta x/2} + \left(\tilde{W}_{ij} - \frac{D_z \tilde{\nu}_T}{\Delta z}\right) \frac{D_z}{\Delta z} - (\nu + \tilde{\nu}_T) \frac{D_z^2}{(\Delta z)^2}\right] U_{ij}^{n+1} = \left[\frac{\tilde{U}_{ij}}{\Delta x/2} - \left(\tilde{V}_{ij} - \frac{D_y \tilde{\nu}_T}{\Delta y}\right) \frac{D_y}{\Delta y} + (\nu + \tilde{\nu}_T) \frac{D_y^2}{(\Delta y)^2}\right] U_{ij}^{n+1/2} + f_x^{n+1/2} - \frac{p_{ij}^{n+1} - p_{ij}^n}{\rho \Delta x} \tag{18b}$$

Equations (18) can be efficiently solved using a tri-diagonal matrix solver due to the banded nature of the differentiation stencils. Similar two-step, discretized equations can be written for (1c-1g) and (7), noting that for the pressure solve, the steps in  $\Delta x$  are replaced with steps in  $\Delta \tau$ .

To calculate a consistent solution for all flow variables, an iterative approach is used at every downstream position. As outlined in algorithm 1, starting from a known solution at  $x_n$  each of the governing equations are solved in sequence for the next values of  $U^{n+1}$ ,  $V^{n+1}$ ,  $W^{n+1}$ ,  $K^{n+1}$ ,  $K^{n+1}$ ,  $K^{n+1}$ , and  $K^{n+1}$ . These solutions are repeated until the difference between successive iterations converge below a predefined tolerance.

The simulations presented below typically used grid sizes of  $(N_y,N_z)=(81,41)$  with mesh sizes of  $\Delta y \times \Delta z=10 \text{m} \times 10 \text{m}$  for the IEA 15 MW reference turbine. Initial refinement studies indicated that streamwise step-sizes of  $\Delta x/R=0.5$  - 1 were possible using this formulation, where R=120m is the turbine-radius of the IEA 15 MW, and provided a good balance between accuracy and computational efficiency. For the single turbine runs discussed in Sec. 4, the simulations required between 10-25 seconds on 1 Intel Xeon Platinum 8480+ CPU.

# 2.4 Actuator disk turbine model




The parabolic formulation of the RANS momentum equations (1b-1d) provides a natural means to represent the wind turbine in the computational domain. Similar to other high fidelity wind turbine simulation codes (see Sec. 3.1), the turbine rotor forces can be included as body forces in the momentum equations through an actuator disk model. Multiple choices of actuator disk models are available in the literature, including the Joukowsky actuator disk model (Sørensen et al., 2020), but the current study uses the uniformly loaded actuator disk model due to its simplicity. This model computes the rotor disk forces based on

# Algorithm 1 To advance the RANS solution to $x_{n+1}$ from $x_n$

```
Set iteration counter m=0  \begin{aligned} &\text{Set } \psi_{ij}^{n+1,m} = \psi_{ij}^n \text{ for } \psi = U, V, W, k, \epsilon, \Theta, p \\ &\text{while } |\psi_{ij}^{n+1,m} - \psi_{ij}^{n+1,m-1}| > \varepsilon_{TOL} \text{ do} \\ &\text{m=m+1} \\ &\text{Calculate } U_{ij}^{n+1,m} \text{ from } U_{ij}^n \text{ using equation (18)} \\ &\text{Calculate } V_{ij}^{n+1,m} \text{ from } V_{ij}^n \\ &\text{Calculate } W_{ij}^{n+1,m} \text{ from } W_{ij}^n \\ &\text{Calculate } k_{ij}^{n+1,m} \text{ from } k_{ij}^n \\ &\text{Calculate } \epsilon_{ij}^{n+1,m} \text{ from } \epsilon_{ij}^n \\ &\text{Calculate } \Theta_{ij}^{n+1,m} \text{ from } \Theta_{ij}^n \\ &\text{Calculate } P_{ij}^{n+1,m} \text{ from } P_{ij}^n \\ &\text{Calculate } p_{ij}^{n+1,m} \text{ from } p_{ij}^n \end{aligned}
```

the density  $\rho$ , thrust coefficient  $C_t$ , freestream velocity  $U_{\infty}$ , and rotor normal  $\hat{\mathbf{n}}_T$ :

$$\mathbf{f}_T = \frac{1}{2} C_t \rho U_\infty^2 g(r) \hat{\mathbf{n}}_T \tag{19}$$

The thrust coefficient is given as a predetermined function of the free-stream wind speed  $C_t = C_t(U_\infty)$ , and  $U_\infty$  is computed using the rotor averaged velocity. In equation (19) the actuator force  $\mathbf{f}_T$  is applied to all points on the rotor disk with the hub location  $(x,y,z) = (x_T,y_T,z_h)$ . A blending function g(r), where  $r = \sqrt{(y-y_T)^2 + (z-z_h)^2}$ , is used to avoid a sharp discontinuity in the applied force at the rotor disk edge. In the current work, the hyperbolic tangent blending function

$$g(r) = \frac{1}{2} \left[ 1 - \tanh\left(\frac{r - R}{r_{\Delta}}\right) \right],\tag{20}$$

is used, where  $r_{\Delta}$  is a smoothing parameter. Note that the actuator force  $\mathbf{f}_T$  is calculated as a force per unit area and is divided by  $\Delta x$  to be included as a body force per unit volume in equations (1b-1d). Multiple turbines and wake steering effects can be captured by superposing multiple actuator turbine forces and adjusting the directions of the rotor normals.

### 2.5 Calibration of the RANS model



The RANS closure model was calibrated by comparing the rotor plane velocity U in the RANS against corresponding planes from LES for the cases listed in table 1 to be described in more detail later. Similar to the approach in Cheung et al. (2025), the calibration only compares the rotor planes at a distance of 4D and 6D downstream of the turbine since the RANS model does not account for the hub and nacelle regions present in the LES. The cost function for the calibration is the  $\mathcal{L}_2$  norm of the difference in the streamwise velocity between the LES and the RANS rotor planes. The extent of the rotor plane for the calibration are  $y \in [-1.5D, 1.5D]$  from the center of the turbine disk and  $z \in [0,400m]$ , discretized into a grid of uniformly spaced [50,21] points. The parameters for calibration were picked to be the coefficients,  $C_{\mu}, C_{1\varepsilon}, C_{2\varepsilon}$ , of the k- $\epsilon$  RANS closure model and the coefficient  $C_k$  for the inflow boundary condition in equation 13. The L-BFGS-B (Byrd et al., 1995; Zhu

https://doi.org/10.5194/wes-2025-249 Preprint. Discussion started: 24 November 2025

© Author(s) 2025. CC BY 4.0 License.



et al., 1997) algorithm as implemented in scipy was used for the calibration. The optimal values from this calibration were  $C_{\mu}=0.076, C_{1\varepsilon}=1.46, C_{2\varepsilon}=1.92$ , and  $C_k=0.72$ . Comparisons of calibrated RANS results in Fig. 3, 4, and 6 show qualitative agreement between the wake velocity planes from the calibration cases, especially at x=4D and x=6D. As further validation, Fig. 5 shows great qualitative agreement between TKE in the RANS and the LES calculations, noting that TKE was not used as part of the calibration calculations. These results are further discussed in Sec. 4.

# 3 Simulation comparison details

In the following sections, we provide details on the high-fidelity LES methodology used for calibration and comparisons of the RANS models. The equivalent semi-empirical FLORIS wake models are also discussed, and information on the atmospheric conditions and turbine configuration used in this study are included in Sec. 3.3.

#### 210 3.1 AMR-Wind code description

Following an approach similar to that described in Cheung et al. (2025), LES data were collected for calibration and comparison purposes by performing simulations with AMR-Wind (Sharma et al., 2024; Sprague et al., 2020; Kuhn et al., 2025), a massively parallel, block-structured adaptive-mesh solver designed for simulating wind turbines and wind farms. AMR-Wind solves the incompressible and low Mach number formulations of the Navier-Stokes equations with transport equations for temperature, subgrid-scale kinetic energy, and additional scalars required for wind farm LES. The spatial discretization employs a second-order finite volume method, coupled with a second-order temporal integration scheme. AMR-Wind includes comprehensive atmospheric boundary layer (ABL) physics modules: ABL forcing, Boussinesq buoyancy, Coriolis effects, body forcing to preserve precursor-derived inflow conditions under turbine blockage. It also includes forcing terms from an actuator line turbine representation (following implementations in Brown et al. (2025) and Hsieh et al. (2025)) that is derived from coupling to OpenFAST (Jonkman et al., 2018; NREL, 2023; Brown et al., 2024) (following implementations in Brown et al. (2025) and Hsieh et al. (2025)). The framework leverages AMReX for data structures, parallelism abstractions, and performance portability across heterogeneous computing architectures (Zhang et al., 2019), demonstrating robust performance across diverse systems and applications (Fedeli et al., 2022; Henry de Frahan et al., 2022, 2024).

# 3.2 FLORIS model description

The RANS model is compared to a steady-state engineering model using the FLOw Redirection and Induction in Steady-state (FLORIS) tool (NREL, 2025). FLORIS is a widely-used wind farm simulation software designed for wind farm layout and control optimization that can predict the time-averaged three-dimensional flow field and turbine power of a wind farm. Following Yalla et al. (2025), the empirical Gaussian model in FLORIS is used here to represent the steady-state wakes. In this model, the normalized wake velocity deficit,  $u/U_{\infty}$ , is represented by a Gaussian centered on lateral and vertical wake centers,

 $\delta_{y}$  and  $\delta_{z}$ , as





$$u/U_{\infty}(x) = 1 - C \exp\left(-\frac{(y - \delta_y(x))^2}{2\sigma_y(x)^2} - \frac{(z - \delta_z(x))^2}{2\sigma_z(x)^2}\right). \tag{21}$$

The standard deviations,  $\sigma_{y,z}(x)$ , represent the wake widths as a function of streamwise distance, x, and are modeled as

$$\sigma_{y,z}(x) = \int_{0}^{x} \sum_{i=0}^{n} k_i \mathbf{1}_{[b_i,b_{i+1}]}(x') dx' + M_j(x) dx' + \sigma_{y_0,z_0},$$
(22)

which depend on several adjustable parameters, including a constant initial wake width,  $\sigma_{y_0,z_0}$ , and a set of parameters,  $k_i$ , that control the wake expansion rate between break-point locations  $b_i$  and  $b_{i+1}$ . For each turbine (indexed by j), the wake widths also include a mixing term,  $M_j$ , that represents the effects of atmospheric turbulence intensity (TI) and wake overlap on wake spreading as

$$M_j(x) = \omega_v \sqrt{\left[\sum_{\substack{i=1\\i\neq j}}^{N_{turb}} \left(\frac{\Omega_{ij}a_i}{((x_j - x_i)/D_i)^2}\right)^2 + (\gamma I)^2\right]},$$
(23)

where  $\Omega_{ij}$  quantifies the area of overlap of the wake of turbine i onto turbine j,  $a_i$  is the axial induction factor of the ith turbine, and I is the turbulence intensity. The parameters  $\omega_v$  and  $\gamma$  can be adjusted to control the strength of the wake mixing term. 240

In Yalla et al. (2025), the empirical Gaussian model in FLORIS was calibrated using LES data from a  $3 \times 3$  array of IEA 15 MW turbines operating in one of the stable wind conditions considered in this study, specifically the Med WS case described in Sec. 3.3. Therefore, the calibrated empirical Gaussian parameters from Yalla et al. (2025) are directly applied here to compare FLORIS with the LES and RANS models. It is important to note that Yalla et al. (2025) focused on estimating annual energy production (AEP) for different wind farm flow control strategies, and therefore calibrated the empirical Gaussian parameters based on turbine power, rather than wake quantities of interest which are the focus here.

#### 3.3 **Simulation cases**

The comparisons between the RANS model, high fidelity LES, and FLORIS calculations were done using two stably-stratified ABL conditions. These conditions where previously studied by Frederik et al. (2025), Brown et al. (2025), and Cheung et al. (2025), and contain the necessary shear, veer, and stratification effects for evaluating the accuracy of the parabolic RANS model. As described in Brown et al. (2025), the offshore ABL conditions are derived from floating buoy lidar measurements taken near the coast of the New York Bight. Two representative low-TI, stable conditions with wind speeds below rated were selected for this study (table 1). In AMR-Wind, the precursor ABL simulations were generated by imposing negative surface ground temperature rates and adjusting the surface roughness  $z_0$  until the horizontally averaged ABL profiles matched the desired targets. Similarly, the surface heat flux  $q_w$  and the surface roughness  $z_0$  were adjusted in the RANS inflow conditions to match the measured lidar profiles.

A comparison of the AMR-Wind LES, RANS, and buoy lidar profiles is shown in Fig. 2. For both the Low WS and Med WS cases, close agreement is observed between the AMR-Wind and RANS horizontal velocity  $U_h$  profiles, as well as with the

**Table 1.** Hub-height wind speed conditions used in the turbine wake study. All values are taken from the simulated atmospheric boundary layer as described in Sec. 3.3.

| Name   | Wind-Speed (WS) | Turb. intensity (TI) | Shear Exponent | Rotor disk veer |
|--------|-----------------|----------------------|----------------|-----------------|
| Low WS | 6.52 m/s        | 0.036                | 0.142          | 7.9°            |
| Med WS | 9.05 m/s        | 0.031                | 0.160          | 8.9°            |

Figure 2. Inflow comparison between the RANS model, AMR-Wind LES, and the floating buoy lidar data for the horizontal wind speed  $U_h(z)$  and veer  $\theta(z)$  profiles for the two ABL conditions in table 1.

lidar measurements. Fig. 2 also shows that the linear veer profile  $\theta(z)$  used in the RANS inflow also matches the AMR-Wind LES veer profile over the rotor disk.

The offshore IEA 15 MW reference turbine is used for all wake comparisons in this study. The major characteristics of the IEA 15 MW turbine are given in table 2, with additional details provided by Gaertner et al. (2020). In the AMR-Wind LES simulations, the OpenFAST actuator line representation of the IEA 15 MW turbine is used with the open source ROSCO (NREL, 2021) wind turbine controller. For the parabolic RANS and FLORIS model, the variation of the thrust coefficient  $C_t$  with wind speed is specified to match the operating curve of the IEA 15 MW design.

**Table 2.** Details of the IEA 15 MW reference turbine

| Turbine Parameter  | Value     |  |
|--------------------|-----------|--|
| Hub height         | 150 m     |  |
| Rotor diameter $D$ | 240 m     |  |
| Rated wind speed   | 10.59 m/s |  |
| Design $C_t$       | 0.804     |  |
| Design TSR         | 9.0       |  |

# 4 Wind turbine wake comparisons

Comparisons of wake behavior between the SANDWake3D RANS model, AMR-Wind LES, and FLORIS calculations are discussed in the following sections. The results for a single turbine wake in the Low WS and Med WS ABL conditions are considered first in Sec. 4.1, before examining the two turbine case in Sec. 4.2. Lastly, the RANS model is demonstrated on a nine turbine wind farm configuration in Sec. 4.3.

# 4.1 Single turbine cases





A qualitative view of the wake evolution for the single turbine configuration in the Low WS ABL condition is provided in Fig. 3. In this figure, the steady streamwise velocity for the LES, RANS, and FLORIS models are displayed at various rotor planes at different downstream distances ranging from x=2D to x=8D. From these plots, several observations can be made regarding the choice of wake model on the predicted wake behavior. When comparing the LES solutions against the RANS in the far wake, for x>5D, we see a similar degree of wake skew and stretching due to the ambient veer in the ABL. In the near wake region, some differences in the centerline wake deficit can be seen, and this can be attributed to the difference between the uniformly loaded disk model in RANS and the actuator line model in AMR-Wind. The latter model includes a nacelle and hub drag model and also captures the variation in loading near the blade root sections. This leads to lower centerline wake deficits compared to the RANS model immediately downstream of the rotor at x=2D, although this difference is less apparent by x=4D.

Compared to the LES and RANS results, the FLORIS empirical Gaussian model also generally captures the wake spread and deficit behavior for the single turbine configuration, and also accounts for the ambient shear from the inflow. However, the effects of veer are not directly included in the empirical Gaussian model, which instead reduces the overall wake deficit to account for the effects of veer on the power of downstream turbines. Wake skewing and stretching are also not present in the FLORIS wake results, which remain axisymmetric at all downstream locations by design.

A similar comparisons of the turbine wake behavior for the Med WS condition is shown in Fig. 4, and similar conclusions can be drawn between the LES, RANS, and FLORIS models. Both AMR-Wind and SANDWake3D capture the effects wake skewing and wake stretching, as opposed to the empirical Gaussian model. In the Med WS condition, we also observe a

https://doi.org/10.5194/wes-2025-249 Preprint. Discussion started: 24 November 2025

© Author(s) 2025. CC BY 4.0 License.




290 unique impact of veer on the wake development in the LES and RANS results, where the wake deficits persist much farther downstream at lower elevations compared to higher elevations.

The amount of wake generated turbulence in the single turbine cases can be examined in both the LES and RANS models. While the resolved TKE in the AMR-Wind calculations and the modeled TKE in the RANS model are not directly equivalent quantities, the two can provide some indication for the degree of mixing happening inside the wake. Contours of the TKE for both the Low WS and Med WS cases are shown in Fig. 5 at different distances downstream. As expected, the TKE distribution generally aligns with the regions of wake shear, and the overall magnitude and evolution of TKE in the RANS model agrees with the results from the LES calculations. Note that in these cases with both shear and veer, the TKE is less heavily concentrated near the lower surface, which might explain the persistence of the wake deficit at lower elevations.

A more quantitative assessment of the RANS wake model is presented in Fig. 6 for both the Low WS and Med WS conditions. In those figures, the hub-height velocity and TKE profiles are plotted for both the LES and RANS solutions. Downstream of the near wake region, the normalized velocity profiles show good agreement between the two solution methods. While the peak values of TKE profile are underestimated in the RANS model, the overall magnitude and distribution of the wake added turbulence is well captured by SANDWake3D.

#### 4.2 Two turbine case

Additional simulations were carried using the SANDWake3D model on a two turbine configuration and evaluated against the counterpart AMR-Wind calculations. In this case, a second IEA 15 MW reference turbine was placed 5D downstream of the first turbine in the Med WS ABL condition. This matches the two turbine configuration studied in previous works (Frederik et al., 2025), and allows the accuracy of the wake and turbulence superposition capabilities of the parabolic RANS model to be assessed against higher fidelity models.

Hub-height contours of the time-averaged velocity and TKE between the AMR-Wind LES and SANDWake3D RANS model are shown in Fig. 7. Due to the parabolic nature of the problem, the RANS results upstream of x=5D remain unchanged, but the inclusion of the second turbine still resulted in very favorable comparisons with the LES without any additional adjustments of the calibration coefficients or changes to the boundary conditions. From the hub-height velocity contour comparisons, the qualitative behavior of the wake deficit and wake spread of the downstream turbine wake matches the LES calculations. Similarly, the distribution and magnitude of TKE in the downstream wake predicted by the RANS model generally matches the resolved TKE computed by AMR-Wind.

A more quantitative comparison of the hub-height velocities and TKE is provided in Fig. 8. Of particular interest are the hub-height and velocity profiles close to the second turbine location, and farther downstream in the second wake. Within the first diameter of the second rotor, at x = 5D - 6D, both the velocity and TKE profile are well captured, and the centerline differences immediately downstream of nacelle region are not as pronounced as in the single turbine comparisons. The increase in the RANS wake added turbulence at x = 6D due to the presence of the second turbine also agrees well with the LES calculations. Farther downstream, the RANS wake profiles continue to show good agreement with the LES profiles. However,

Figure 3. Comparison of the streamwise velocity u(y,z) for the Low WS inflow condition, as computed by AMR-Wind LES, SANDWake3D RANS, and FLORIS empirical Gaussian methods. Contours of u(y,z) are plotted with units of meters/second at distances x/D=2,4,6,8 downstream of the turbine. The dashed circle corresponds to the location of the rotor disk of the IEA 15MW reference turbine.

Figure 4. Comparison of the streamwise velocity u(y,z) for the Med WS inflow condition, as computed by AMR-Wind LES, SANDWake3D RANS, and FLORIS empirical Gaussian methods. Contours of u(y,z) are plotted with units of meters/second at distances x/D=2,4,6,8 downstream of the turbine. The dashed circle corresponds to the location of the rotor disk of the IEA 15MW reference turbine.

Figure 5. Comparison of the turbulent kinetic energy for the Low and Med WS inflow condition, as computed by AMR-Wind LES and SANDWake3D RANS methods. Contours of TKE are plotted with units of  $m^2/s^2$  at distances x/D=2,4,6,8 downstream of the turbine. The dashed circle corresponds to the location of the rotor disk of the IEA 15MW reference turbine.



**Figure 6.** Hub-height profiles of the normalized velocity and TKE for the single turbine wake in the Low WS and Med WS ABL conditions, as computed by the AMR-Wind LES and SANDWake3D RANS codes.

the RANS TKE profiles at x=8D-9D under-predict the peak magnitude of wake-added turbulence, although the general distribution still qualitatively agrees.

The three-dimensional nature of the downstream wake evolution is shown in Fig. 9. Within the first diameter downstream of the second turbine, at x = 5D - 6D, the interaction of the second turbine wake with the skewed wake from the first turbine is well represented using the current parabolic RANS approach. Farther downstream at x = 8D - 9D, the eventual merging of both wakes into a single skewed wake is also consistent between the LES and RANS models. From Fig. 9, the evolution of TKE in the second wake using the parabolic RANS model also matches the observed TKE distribution from the AMR-Wind LES results, although the peak turbulence values are stronger in the LES.

**Figure 7.** The streamwise velocity (top) and TKE (bottom) on the hub-height plane for the two-turbine configuration. Note that all streamwise distances are measured from the upstream turbine at x/D=0, and the second turbine is located at x/D=5.

Figure 8. Hub-height profiles of the normalized velocity and TKE for the two-turbine configuration in the Med WS ABL conditions, as computed by the AMR-Wind LES and SANDWake3D RANS codes. Note that all streamwise distances are measured from the upstream turbine at x/D = 0, and the second turbine is located at x/D = 5.

Figure 9. Rotor plane contours of the streamwise velocity and TKE for the two-turbine configuration in the Med WS inflow. Note that all streamwise distances are measured from the upstream turbine at x/D=0, and the second turbine is located at x/D=5.

**Figure 10.** The streamwise velocity on the hub-height plane for the 3 row turbine wind farm configuration in the Med WS inflow, as computed by AMR-Wind LES and SANDWake3D RANS methods. The 9 IEA 15 MW turbines are spaced 5D apart in both the lateral and streamwise directions.

# 4.3 Wind farm case



In the last demonstration of the RANS model's capabilities, we simulated a 9 turbine wind farm configuration using both SANDWake3D and AMR-Wind. This configuration matches a case studied by Yalla et al. (2025) and involves a three row wind farm in the 9 m/s Med WS inflow conditions, with IEA 15 MW reference turbines spaced 5D in both the lateral and streamwise directions. The full LES domain was  $10 \text{ km} \times 10 \text{ km}$ , while the RANS domain was approximately  $4 \text{ km} \times 4 \text{ km}$ . The computational expense of the simulations was approximately 86,400 GPU-hrs and 4 CPU-mins, respectively, for the LES and RANS methods.

A qualitative comparison of the solutions is provided in Fig. 10 and Fig. 11. From the hub-height comparisons of the streamwise velocity in Fig. 10, we see that the general wake spread and wake deficit magnitudes are captured by the RANS model. The effects of veer on the second and third row wakes are shown in Fig. 11, and the behavior is consistent with the earlier observations in Sec. 4.1 and 4.2. Downstream of the third row, the wake skew and stretching in both codes appears to evolve slower compared the wake from the first turbine row. Additional work is ongoing regarding the study of the RANS modeled wakes in complex wind farm configurations, and the results will be reported in future studies.

Figure 11. Rotor plane contours of the streamwise velocity, in m/s, for the 9 turbine wind farm configuration in the Med WS inflow. Note that all streamwise distances are measured from the first turbine row at x/D = 0, the second turbine row is located at x/D = 5, and the third at x/D = 10. Note that  $L_y$  is the lateral coordinate measured from the center turbine.

© Author(s) 2025. CC BY 4.0 License.

**Table 3.** Comparison of computational cost for the simulation of a single turbine wake. Note that the AMR-Wind simulations were performed on the Frontier Exascale supercomputer with 1600 GPUs.

| Code       | Finest grid resolution $(\Delta x \times \Delta y \times \Delta z)$ | Time to solution (Wall time) | Computational resources |
|------------|---------------------------------------------------------------------|------------------------------|-------------------------|
| AMR-Wind   | $2.5~\text{m} \times 2.5~\text{m} \times 2.5~\text{m}$              | 17.5 hr                      | 28000 GPU-hr            |
| SANDWake3D | $120~\text{m}\times10~\text{m}\times10~\text{m}$                    | 10-25 s                      | 




model described in Sec. 2.4 can also be replaced with other actuator disk models, such as the Joukowsky disk model (Sørensen et al., 2020). This would allow interactions between veer and swirl to be included in future wake simulations. Similarly, the effects of yaw misalignment and wake steering on wake behavior is also naturally included in this formulation and can be the subject of future studies. Lastly, to model the behavior of active wake mixing controls in turbine wakes, a linear stability model can be incorporated into SANDWake3D, similar to the approach of Cheung et al. (2025).

Code and data availability. The AMR-Wind code used for this study is available at https://github.com/Exawind/amr-wind/, and the SANDWake3D code will be made publicly available on Github after the appropriate approvals. The datasets used in this study are available upon request.

Author contributions. LC was responsible for developing the mathematical formulation, model implementation, and manuscript preparation. PM was responsible for calibration of the RANS model coefficients, performance optimization of the RANS model solver, and manuscript contributions. MTHdF was responsible for performance optimization of the SANDWake3D model solver, discussions surrounding the AMR-Wind solver, and manuscript contributions. GY was responsible for the formulation of the RANS model, FLORIS model comparisons, generation of LES data, and manuscript preparations. AH assisted with data post-processing and the comparison of results. KB was responsible for conceptualization, performing portions of the LES, and manuscript review. NdV was responsible for the formulation and development of the RANS model. SKM was responsible for model implementation and manuscript review. MD assisted with editing and review of the manuscript and was also responsible for project organization. MS assisted with editing and was responsible for funding and computer time on OLCF resources.

Competing interests. The authors declare that they have no conflict of interest.

*Disclaimer.* Any subjective views or opinions that might be expressed in the written work do not necessarily represent the views of the U.S. Government. The publisher acknowledges that the U.S. Government retains a non-exclusive, paid-up, irrevocable, world-wide license to publish or reproduce the published form of this written work or allow others to do so, for U.S. Government purposes. The DOE will provide public access to results of federally sponsored research in accordance with the DOE Public Access Plan.

Acknowledgements. Sandia National Laboratories is a multimission laboratory managed and operated by National Technology & Engineering Solutions of Sandia, LLC, a wholly owned subsidiary of Honeywell International Inc., for the U.S. Department of Energy's National Nuclear Security Administration under contract DE-NA0003525.

This work was authored in part by NREL for the U.S. Department of Energy (DOE), operated under Contract No. DE-AC36-08GO28308. This research used resources of the Oak Ridge Leadership Computing Facility at the Oak Ridge National Laboratory, which is supported

https://doi.org/10.5194/wes-2025-249 Preprint. Discussion started: 24 November 2025 © Author(s) 2025. CC BY 4.0 License.

by the Office of Science of the U.S. Department of Energy under Contract No. DE-AC05-00OR22725, and under the ALCC allocation "Grand-challenge predictive wind farm simulations".

This material is based upon work supported by the U.S. Department of Energy, Office of Science, Advanced Scientific Computing Research and Biological and Environmental Research programs through the FLOWMAS Energy Earthshot Research Center.

This research has been supported in part by the Wind Energy Technologies Office within the Office of Energy Efficiency and Renewable
Energy. The views expressed in the article do not necessarily represent the views of the U.S. DOE or the U.S. Government.

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
