# Peer review of "SANDWake3D: A 3D parabolic RANS solver for atmospheric boundary layers and turbine wakes"

_Wind Energy Science, 2025_

## Referee Comment (RC1)

**SANDWake3D: A 3D parabolic RANS solver for atmospheric boundary layers and turbine wakes by Lawrence Cheung et al.**

Reviewer: M. Paul van der Laan, DTU Wind and Energy Systems

December 9, 2025

The authors propose a parabolic Reynolds-averaged Navier-Stokes (RANS) wind farm flow model using two-equation turbulence model including effects of stability and wind veer. The stability is modeled using Monin-Obukhov Similarity Theory (MOST). The model is calibrated against large-eddy simulations (LES) for two similar single wake cases. A double wind turbine and small farm case is used to validate the model against LES.

The article is relatively well written and follows a good structure. The development of fast wake models including atmospheric physics as stability and wind veer, is very relevant for the wind energy community. My main concern with the present work is the limited calibration and model validation in terms of range of thrust coefficients, ambient inflow turbulence intensity and atmospheric stability. Furthermore, the model has several components that are either inconsistent or require more information. I have listed more detailed comments below.

**Main comments**

1. Introduction, Line 45: The authors write *In the WakeBlaster model, a single streamwise momentum equation is solved by advancing 2D planes of the velocity field, and the introduction of wakes is accomplished through direct manipulation of the velocity profiles. However this limits the ability of the model to handle veered or stratified inflow conditions..* I agree regarding the wind veer, but a stratified inflow following MOST could still be handled by a single streamwise momentum equation, as the Wake Blaster model employs. I recommend the authors to rewrite this statement.

2. Section 2.1: The authors write: *Previous studies have shown that the k epsilon model can accurately simulate stratified ABL conditions (Alinot and Masson, 2005) and was also successfully used in prior simplified models for wake dynamics (Cheung et al., 2025)..* However, Alinot and Masson (2005) applied MOST to the k-epsilon model, which represents the atmospheric surface layer, not the atmospheric boundary layer.

3. Section 2.1: The performance of the Durbin time-scale limiter for the application of wind turbine wakes is quite sensitive to the ambient turbulence intensity, as shown in van der Laan and Andersen [2018]. Hence, a general calibration of the model with LES for a wide range of cases may be difficult.

4. How is the TI defined in Table 1? Is based on the TKE? Please clarify as different TI definitions are used in the wind energy community.

5. What is the domain height in the RANS model? Is it 300 m?

6. There are several undefined parameters. What is are the values of the roughness length, Obukhov length, friction velocity, wall temperature, heat flux, and specified lapse rate for the RANS model, for both inflow cases? What is the chosen value of the von Kármán constant?

7. Section 2.2: What are the lateral ($y = 0$ and $y = L_y$) and outflow boundary conditions ($x = L_x$)?

8. Section 2.4: The use of a uniform thrust load distribution typically leads to an overestimated velocity deficit compared to a more realistic force distribution (as for example the model from Sørensen et al. [2020]). This is because a realistic force distribution has a low force near the blade root leading an additional wake recovery, which is not captured by a uniform load distribution, see for example Simisiroglou et al. [2017].

9. Section 2.5: The calibration involves two single wake cases from LES that differ in wind speed (6.52 and 9.05 m/s). However, these inflow wind speeds correspond to nearly the same thrust coefficient, namely, 0.83 and 0.8. Furthermore, a similar ambient inflow TI of 3.1-3.6% is used. Therefore, the calibration is covering a very limited space. I recommend to perform a calibration with LES singles wakes for a range of thrust coefficients and ambient TI levels; e.g. CT=0.5-0.8, TI=0.04-0.15 (corresponding to offshore and onshore conditions). The author could also consider to use different stability conditions/ wind veer in the calibration as the focus of the present work is the inclusion of more complex atmospheric physics.

10. The title includes atmospheric boundary layer, but MOST is used as inflow which represent the atmospheric surface layer. Therefore, I recommend to use the latter instead.

11. There are several model choices that are inconsistent with the chosen RANS equations. This means that the inflow is expected to develop downstream due to an imbalance with the inflow, which is based on my experience with elliptic RANS models:

    (a) It is unclear how the wind veer is modeled in the proposed RANS formulation, as the inflow model is based on MOST, which does not consider wind veer. A Coriolis force is neither considered in the momentum equation. Do the authors directly use the wind veer from the LES model to rotate the MOST profiles? If this is the case, then the applied RANS equations are not balance with the inflow model. For elliptic 3D RANS models, it is common to perform a 1D precursor simulation (height is the only spatial variable) to determine the inflow for the wind farm successor simulation.

    (b) Equation 13: Why is $C_k$ used here? The MOST $k$ profile is normally defined as: $u_*^2/\sqrt{C_\mu}\sqrt{\phi_m/\phi_\varepsilon}$. Hence $C_k$ is related to $C_\mu$ : $5.48C_k = 1/\sqrt{C_\mu}$ and is not an independent parameter. If one changes $C_k$, then one sets effectively a different $C_\mu$. In addition, one other turbulence model constant needs to be adjusted to make the RANS + k-epsilon model in balance with a MOST, see for example Richards and Hoxey [1993]. Following the calibration from Sect. 2.5, the effective eddy viscosity coefficient is $C_\mu = (1/(5.48C_k))^2 = (1/(5.48 \times 0.72))^2 = 0.0642$, which is not equal to the reported value of 0.076, and means that the RANS model is not in balance with the inflow.

    (c) The MOST model of Alinot and Masson (2005) is not in full balance with the RANS equations as the parametrized turbulence model constant (Eq. 5) is only valid up to a certain height, see van der Laan et al. [2017]. Hence, if one has a large vertical domain, then this model is no longer valid.

    (d) The use of an active temperature equation in combination with a nonlinear temperature profile will not result in a steady-state solution. Hence, the inflow will develop downstream.

    If the authors decide to keep their model choices, then I recommend to include a result without turbines to show the downstream development of the inflow, preferable for a large horizontal domain in the $x$-direction, e.g. 50 km.

12. Figure 2: How does the RANS inflow temperature profile compare with the LES results for both inflow cases?

13. Figure 5: At first sight the difference between the low and medium WS cases look significant. However, this is because the dimensional TKE is plotted. I would recommend to normalize the TKE ($k$) by the ambient inflow velocities to avoid such confusion. For example, one

could use $\sqrt{2/3k}/U_{\text{inflow}}$, to get a TI-representative value while not mixing the result with the velocity solution. I expect that the results will be similar between the two cases. This also applies to all other figures where the TKE in dimensional form is plotted. In general, I recommend the authors to normalize all velocity and TKE plots by their respective inflow values.

14. The small wind farm case consisting of 9 turbines took 4 CPU min or 4/3600 CPU hours. Assuming one 1 CPU is used, as it is not mentioned that the code is parallel, then an annual energy production calculation takes $4/3600 * 360 * 22 = 8.8$ hours. This makes the model difficult to use for wind farm optimization purposes.

15. The streamwise pressure gradient is included in the streamwise momentum equation (Eq. 1b). Does this mean that the turbine induction is included? In other words, how does the upstream flow of a single wake look like, for example at 1D upstream?

16. I miss a discussion on some of the model limitations. For example, a parabolic RANS model cannot capture blockage and speed up effects. Effects of an ABL height are neither modeled.

**Minor comments**

1. Equation 1d: Brackets seems to be missing at the right hand side terms.

**References**

P. J. Richards and R. P. Hoxey. Appropriate boundary conditions for computational wind engineering models using the $k$-$\varepsilon$ turbulence model. *Journal of Wind Engineering and Industrial Aerodynamics*, 46,47:145–153, 1993. doi: https://doi.org/10.1016/0167-6105(93)90124-7. URL https://www.sciencedirect.com/science/article/pii/0167610593901247.

N. Simisiroglou, S.-P. Breton, and S. Ivanell. Validation of the actuator disc approach using small-scale model wind turbines. *Wind Energy Science*, 2(2):587–601, 2017. doi: 10.5194/wes-2-587-2017. URL https://www.wind-energ-sci.net/2/587/2017/.

J. N. Sørensen, K. Nilsson, S. Ivanell, H. Asmuth, and R. F. Mikkelsen. Analytical body forces in numerical actuator disc model of wind turbines. *Renewable Energy*, 147:2259, 2020. doi: https://doi.org/10.1016/j.renene.2019.09.134.

M P van der Laan and S J Andersen. The turbulence scales of a wind turbine wake: A revisit of extended k-epsilon models. *Journal of Physics: Conference Series*, 1037(7):072001, jun 2018. doi: 10.1088/1742-6596/1037/7/072001. URL https://doi.org/10.1088/1742-6596/1037/7/072001.

M. Paul van der Laan, Mark C. Kelly, and Niels N. Sørensen. A new k-epsilon model consistent with monin–obukhov similarity theory. *Wind Energy*, 20(3):479–489, 2017. doi: https://doi.org/10.1002/we.2017. URL https://onlinelibrary.wiley.com/doi/abs/10.1002/we.2017.